# Willingness and SARS-CoV-2 Vaccination Coverage among Healthcare Workers in China: A Nationwide Study

**DOI:** 10.3390/vaccines9090993

**Published:** 2021-09-06

**Authors:** Chao Wang, Yu Wang, Bingfeng Han, Tian-Shuo Zhao, Bei Liu, Hanyu Liu, Linyi Chen, Mingzhu Xie, Hui Zheng, Sihui Zhang, Jing Zeng, Ning-Hua Huang, Juan Du, Yaqiong Liu, Qing-Bin Lu, Fuqiang Cui

**Affiliations:** 1Department of Laboratorial Science and Technology & Vaccine Research Center, School of Public Health, Peking University, Beijing 100191, China; wchao@bjmu.edu.cn (C.W.); 2011110184@bjmu.edu.cn (Y.W.); hanbingfeng@pku.edu.cn (B.H.); zts2018@pku.edu.cn (T.-S.Z.); 1916387057@bjmu.edu.cn (B.L.); liuhanyuu@bjmu.edu.cn (H.L.); chenlinyi137@pku.edu.cn (L.C.); 1610306142@pku.edu.cn (M.X.); zhenghui@chinacdc.cn (H.Z.); zhangsihui@bjmu.edu.cn (S.Z.); zengjing@bjmu.edu.cn (J.Z.); hnh@bjmu.edu.cn (N.-H.H.); juandu@bjmu.edu.cn (J.D.); liuyaqiong@bjmu.edu.cn (Y.L.); 2Department of Epidemiology and Biostatistics, School of Public Health, Peking University, Beijing 100191, China

**Keywords:** SARS-CoV-2 vaccine, willingness, confidence, coverage, healthcare workers

## Abstract

*Background:* The SARS-CoV-2 vaccine has been widely rolled out globally in the general populations. However, specific data on vaccination confidence, willingness or coverage among health care workers (HCWs) has been less reported. *Methods*: A cross-sectional online survey was conducted to specify the basic data and patterns of vaccination confidence, willingness and coverage among HCWs nationwide. *Results*: In total, 2386 out of 2583 (92.4%) participants were enrolled for analysis, and the rates of confidence in vaccine, professional institutes and government were 75.1%, 85.2% and 85.4%, respectively. The overall vaccination coverage rate was 63.6% which was adjusted as 82.8% for participants under current medical conditions or having contraindications. Confidence in vaccine safety was shown to be the most related factor to willingness among doctors, nurses, medical technicians and hospital administrators, while confidence in vaccine effectiveness as well as trust in government played the key role in formulating public health employees’ willingness. 130 (7.1% of 1833) participants reporting willingness still not been vaccinated regardless of contraindications. Multivariate analysis among willingness participants showed that males, aged over 30 years, public health employees and higher vaccination confidence had significantly higher vaccination rates with ORs (95% confidence intervals) as 1.64 (1.08–2.49), 3.14 (2.14–4.62), 2.43 (1.46–4.04) or 2.31 (1.24–4.33). *Conclusions**:* HCWs’ confidence, willingness and coverage rates to the vaccine were generally at high levels. Heterogeneity among HCWs should be considered for future vaccination promotion strategies. The population’s confidence in vaccination is not only the determinant to their willingness, but also guarantees their actual vaccine uptake.

## 1. Introduction

The novel coronavirus disease 2019 (COVID-19) pandemic has caused huge losses to life and economy [1]. Till now, signs of remission to the threat of COVID-19 have been blurry due to various conditions of epidemic prevention strategies, policies or vaccine coverage between different countries [2]. The COVID-19 epidemic has been well-controlled in China with most of the current confirmed cases as being imported so far [3]. However, this has been achieved on the basis of a large input of national and social resources and could lead to a potential sharp epidemic rebound if the current strict prevention and control measures were to be deregulated [4]. Thus, the deployment and coverage of safe and effective vaccines against COVID-19 is expected to be the best long-term and cost-effective solution to control the pandemic.

These effective measures are often challenged by individuals delaying or refusing vaccination, even though evidence and reports have shown benefits for personal health and social recovery by reducing viral transmission, preventing serious infections and death [5]. It is difficult to control the pandemic if the population’s vaccination rate does not reach a high coverage rate. Since a majority of countries provide vaccines to residents with no charge, this is generally thought to be a result of low vaccine confidence in the targeted population due to the short development duration, less public awareness, poor vaccine knowledge and reported adverse events following immunization, etc. [6,7].

Reports on the willingness to vaccination have been various, indicating 50% to 90% uptake willingness in general populations [8]. A global survey reported 71.5% of participants on average would be very or somewhat likely to take a COVID-19 vaccine and the highest rate of nearly 90% in Chinese population [9]. Several studies also indicated willingness to accept COVID-19 vaccine did not necessarily result to vaccine uptake, showing a large gap that still needs to be closed to improve the vaccination coverage. In the meantime, reports on Chinese population’s willingness to get vaccinated range from 34.8% to 91.3% [7,10,11,12,13,14,15], suggesting that vaccination willingness may be postponed now that a vaccine is available compared to when it was not available [6,10,11,12,13,14].

Up until now, the majority of studies on the population’s willingness to accept COVID-19 vaccination have focused on general population [5]. There has been limited studies on the prevalence of willingness to be vaccinated among health care workers (HCWs) [5] who are one of the nine key populations targeted for early vaccination in the first vaccination campaign that started on December 15, 2020 in China. As a most special group, HCWs are the key social force in the pandemic control and face the highest risk of exposure to SARS-CoV-2 virus [16]. It is critical to ensure the safety of HCWs not only to safeguard continuous disease control and treatment but also to ensure they do not transmit the virus under more frequent social communication conditions [17]. A retrospective analysis in England showed that overall infection level among HCWs was probably underestimated due to asymptomatic seroconversion [18]. In the aspect of COVID-19 vaccination, HCWs play an important role in vaccine propagation and education, and their vaccination willingness can also influence the attitude and uptake choice of the general population [15,19]. HCWs are usually thought to be opinion leaders by general populations in terms of medical issues, even though they might hold misleading or biased attitudes on specific areas [15,19], so it is important and essential to study their willingness and attitude to COVID-19 vaccines to promote the vaccination coverage as a whole.

We conducted this study to investigate the willingness and coverage of COVID-19 vaccination among five kinds of HCWs one month after the beginning of the key population vaccination campaign in China. In addition, since the vaccination has been promoted voluntarily, we wished to explore the relationship between HCWs’ vaccine confidence and vaccination willingness and detect the unvaccinated coverage of people willing to be vaccinated, offering data support in formulating immunization strategies and promoting the population’s COVID-19 vaccine coverage.

## 2. Methods

### 2.1. Study Design

A cross-sectional online survey was conducted basing on an online platform from 10 January to 22 January 2021. It was an open online survey targeting Chinese residents aged 18 to 60 years. People willing to participate could complete the questionnaire, which was offer through mainstream media like Wechat and Weibo. The questionnaire could be completed in around three minutes either by mobile phone or computer. Data was collected on WenJuanXing, an online platform providing which provides online questionnaire design and survey functions equivalent to those of Amazon Mechanical Turk, Qualtrics (Seattle, WA, USA), SurveyMonkey (San Mateo, CA, USA) or CloudResearch. The HCWs’ data was extracted from the total population database.

### 2.2. Data Collection

The structured questionnaire contained information on demographic characteristics including gender, age, education, residence, and five classes of HCWs (doctors, nurses, public health employees, medical technicians and hospital administrators) were studied. Information on vaccine confidence and willingness to accept the COVID-19 vaccination were simultaneously collected, which were detailed in our previous report based on the whole database, as well as the quality control process [6].

The sample size was calculated at least 855 with 171 for each type of HCWs by setting willingness proportion among HCWs as 70% through the cross-sectional study equation, in which α was set as 0.05 and the power was set as 90%. We conducted a post hoc analysis to evaluate the values of the collected samples for the exploration of our study purposes.

### 2.3. Statistical Analysis

A chi-square test was used to test the differences in the proportion of vaccine confidence, willingness and vaccination uptake among the different demographic groups. Logistic regression models were applied to the total sample and each HCW sub-group separately to detect the different patterns of the relationship between vaccine confidence (X) and willingness (Y). Demographic characteristics were introduced into logistic regression model as covariates. Odds ratio (OR) and 95% confidence interval (CI) were used for detection of the strength of association between study variables.

Participants were further categorized into willing & vaccinated, willing & unvaccinated, unwilling & vaccinated, or unwilling & unvaccinated groups to explore the factors relating to lack of vaccination among the willing population. A logistic regression model was further applied among participants that reported willingness to get vaccinated to explore the effects of vaccine confidence on actual vaccination uptake. The odds ratio (OR) and 95% confidence interval (CI) were also calculated and displayed. SPSS (version 22.0, IBM, Armonk, NY, USA) and R statistical software (version 3.6.3; appendix p 11) were used for cleaning up the data and statistical analysis. The significance level was considered when the *p* value was less than 0.05.

### 2.4. Ethical Approval

This study was approved by Peking University Institutional Review Board (IRB00001052-21001); exemption for informed consent was granted.

## 3. Results

### 3.1. Overview of Participants’ Characters

In total, 2583 medical employees were extracted out of 9531 total participants in this online survey. Among all the HCWs, 167 were excluded due to duplicated IP addresses or because they resided outside mainland China, 24 more were excluded due to the quality control question, and 6 were excluded due to logic errors. Finally, 2386 valid questionnaires were recruited for data analysis. The rate of completion was 92.4% (2386/2583), which was deemed sufficient for the calculated sample size. The samples enrolled for data analysis covered 31 provinces in China among which Shandong, Jiangsu, Qinghai and Beijing offered the largest number of participants, while participants belonged to Heilongjiang and Yunnan were of the least abundant. The proportion of participants from the east, middle and west regions of China were 41.3%, 20.3% and 38.4%, respectively.

Among the 2386 HCWs, 1502 (63.0%) were female compared to the 71.8% national figure; 1411 (59.1%) were aged below 40 years which was similar to the overall national proportion (59.1%); 1684 (70.6% vs. 69.9% nationally) had a bachelor’s degree; participants living in urban areas accounted for 76.3%, which is higher than the 50.9% national proportion (Table 1 and Appendix A). The power of post hoc analysis was satisfactory enough for data analysis (Appendix A). In summary, 75.1% (1792) participants had confidence in COVID-19 vaccine, and 1929 (80.8%) participants reported willingness to accept COVID-19 vaccination. Males, aged over 40 years, with lower educational background and rural residents showed a significantly higher willingness. The overall vaccination coverage rate was detected as 63.6% (1518/2386) which was adjusted as 82.8% (1518/1833) for participants under current medical conditions or having contraindications (Figure 1A and Appendix A).

### 3.2. Analysis by Subgroups of HCWs

Among 603 doctors, 54.6% of the participant were females, 34.7% (209/603) were between 30 to 39 years old and 71.5% were from an urban area. Doctors showed the highest education level as 32.2% of them had a master’s degree or above (Table 1). Their confidence rates in the vaccine, professional institute or the government were reported as 71.3%, 82.3% or 82.4%, respectively (Figure 2 and Appendix A). More specifically, they showed relatively low confidence in vaccine safety (69.2%) and effectiveness (68.7%) compared to other dimensions of vaccine confidence (Appendix A). 79.4% of them reported their intention to get vaccinated against COVID-19 (Figure 2 and Appendix A), and confidence in vaccine safety and importance was shown by multivariate analysis to be the most relevant issues regarding their willingness (Figure 3B and Appendix A). Their current rate of vaccination was 56.2%, which was be increased by 79.1% after adjusting for participants with current medical conditions or contraindications (Appendix A).

The majority of nurses were female (92.3), were between 30 to 39 years of age (37.1%), and had a bachelor’s degree (91.7%). 62.6% of them were from urban areas (Table 1). Their confidence in vaccination, professionals or the government was shown to be at a relatively low level as 67.7%, 81.1% or 83.4% compared to other medical employees, as well as their willingness to be vaccinated (74.0%) (Figure 2 and Appendix A). The most influencing factor for their vaccination willingness was vaccine safety (OR 9.39, 95%CI: 4.08–21.56), followed by their confidence in the government (OR 3.25, 95%CI: 1.14–9.22) (Figure 3C and Appendix A). The surveyed and contraindication-adjusted vaccination coverage among them were 57.4% or 78.8%, respectively (Appendix A). 

Public health employees showed similar demographic characteristics as doctors, except for a relatively higher proportion of urban residents (Table 1). They showed the highest level of vaccine confidence (83.3%, 89.4% and 87.6%) and willingness (88.1%) comparing to the other four kinds of HCWs in our study (Figure 2 and Appendix A). Moreover, a higher willingness rate was detected among public health employees that were confident in vaccination (Appendix A). Vaccine importance (OR 2.33, 95%CI: 1.22–4.48), safety (OR 2.21, 95%CI: 1.17–4.15) and effectiveness (OR 5.84, 95%CI: 3.08–11.11) showed significant relationships with their vaccination willingness, while vaccine effectiveness was shown to be the most cited issue (Figure 3D and Appendix A). Their current and contraindication-adjusted vaccination rates were 76.4% or 89.5%, respectively (Appendix A). 

In addition, 65.4% of medical technicians were female. The majority of them were aged below 39 years (68.8%), had a bachelor’s degree or above (94.4%) and lived in an urban area (75.6%) (Table 1). They reported the highest confidence in the government at 89.3% (Figure 2 and Appendix A). Both vaccine importance (OR 4.25, 95%CI 1.64–10.98) and safety (OR 4.25, 95%CI 1.64–11.02) showed a significant relation to their vaccination willingness (Figure 3E and Appendix A). Their current and contraindication-adjusted vaccination rates were 54.7% or 79.0%, respectively (Appendix A).

61.1% of hospital administrators were female. The majority of them were aged below 39 years (61.4%), had a bachelor’s degree or above (90.2%) and lived in an urban area (82.2%) (Table 1). Their confidence in the vaccine, professionals or government was 69.1%, 81.8% or 84.0%, respectively (Figure 2 and Appendix A). Comparing to those without confidence in vaccine safety, participants that trust the safety of vaccination showed the highest OR (5.89, 95%CI 2.57–13.51) regarding vaccination willingness (Figure 3F and Appendix A). Their current and contraindication-adjusted vaccination rates were 52.4% and 72.7%, respectively (Appendix A).

### 3.3. Combined Willingness & Vaccination Status Analysis

We further classified the participants into four subgroups: unwilling & unvaccinated, unwilling & vaccinated, willing & unvaccinated and willing & vaccinated. There were 130 (7.1% of 1833) participants reporting willingness but still not vaccinated, regardless of medical condition or contraindications (Figure 1B). Participants who reported total confidence in the COVID-19 vaccine showed the highest rate of willingness & vaccination (87.4%, 1173/1408), which was much higher than that with no vaccination confidence (32.3%, 62/192). On the other hand, those showing willingness and unvaccinated participants reported a balanced proportion in vaccination confidence. A logistic regression model was further applied to the group of participants that reported vaccination willingness, to detect the factors related to their vaccine uptake. Results showed that ORs of males, aged over 30 rears, public health employees and high vaccination confidence had statistical significance, measured at 1.64 (95% CI 1.08–2.49), 3.14 (95% CI 2.14–4.62), 2.43(1.46–4.04) or 2.31 (95% CI 1.24–4.33), respectively (Figure 1C and Appendix A). 

## 4. Discussions

Through this cross-sectional online survey on SARS-CoV-2 vaccination, we add new specific insights into the prevalence and different patterns of factors that influence vaccine confidence, willingness and coverage among HCWs in China by using nationally comparable survey data. Our study is the first to identify the gap between willingness and actual uptake of the SARS-CoV-2 vaccine at the beginning of its full access for the HCW population.

Vaccination is critical to control COVID-19, which relies on safe and effective vaccines and also on high levels of uptake by the public over time [6]. As a prolonged phenomenon, vaccine hesitancy or lack of confidence in various vaccines has aroused much attention [20]. Some are wondering if the pandemic of COVID-19 would eliminate vaccine hesitancy making population much more confident in the SARS-CoV-2 vaccine [21]. However, studies have indicated that population’s confidence in SARS-CoV-2 vaccine varies across countries/regions, and seems even worse after its utilization [8,10]. According to our results, confidence in the safety and effectiveness of the vaccine was much lower than the confidence in professional institutes or the government, suggesting that nearly one quarter of HCWs didn’t trust the newly developed vaccine as a reliable measure for dealing with the newly emerged pandemic.

However, compared to general population, HCWs’ confidence was still at a higher level [6,22,23], given the nature of the HCW profession [14]. HCWs have a stronger medical background, and are thus more knowledgeable about the safety, effectiveness, and long-term benefits of vaccines [24]. Furthermore, there has been very limited data on COVID-19 vaccination perception among employees of public health. According to our results, they showed the highest proportion of confidence regardless of demographic characteristics, no matter in vaccine, professional institute or the government, which is probably due to the public health attribute of vaccines [25,26,27], and a reflection of the nature that vaccines are more of public health issues in terms of disease prevention. On the contrary, nurses were found to report the lowest confidence. It has been suggested that this may be correlated with their misconceptions regarding the vaccine effectiveness or their knowledge level [28,29], while our result supposed it may be also be due to an imbalance of gender composition since the majority of nurses are females. Other kinds of HCWs like doctors, medical technicians or hospital administrators also showed high level of vaccination confidence. HCWs are not only a key target population for vaccination, but also a trusted source of vaccine information fo the general public [30]. Measures to eliminate the hesitation to vaccinate among HCWs and the lack of knowledge about vaccines have the potential to improve the confidence of the population as a whole [29,30,31].

Given the current concerns about the COVID-19 pandemic, willingness to be vaccinated against COVID-19 has become an important public health issue. 80.8% of HCW participants reported their willingness to be vaccinated in the present survey, which is similar to the data from previous reports in France [32], Italy [33], Israel [34] among HCWs [35], whereas it was much higher than that observed in Congo (27.7%) [36] and Greece (43.3%) [37]. Generally, the surveyed willingness of HCWs is higher than the values observed in general populations across countries/regions, given the number of HCWs infected and even sacrificing their lives in fighting this disease [8]. HCWs also reported higher willingness to receive COVID-19 vaccination comparing to other vaccines, like influenza vaccine (31.1% to 69%) [38,39,40]. However, effective communication and innovative strategies should still be promoted since the studies have highlighted a growing trend of vaccine hesitancy among HCWs [41].

More specifically, the results show different vaccination willingness population patterns. Public health employees were more likely to be willing if they had confidence in vaccination, while administration employees were more likely to be unwilling when they lacked confidence in vaccination. This was also potentially due to the job content and knowledge background, indicating a more frequent suspicious context among administration employees [8]. In this light, measures increasing the vaccination confidence would be more effective in promoting willingness among public health employees but implementation seems to be more urgent among hospital administrators.

Participants who are younger aged females showed significantly lower confidence or willingness than others, which is in line with previous studies [15,42,43]. As newly developed and with short clinical observation, scientific data or proof on the effectiveness or safety of COVID-19 vaccine had not been convincing enough. Females and youngsters are argued to be more susceptible to rumors or misleading information in terms of their frequent exposure to new media and to pay more concern to side effects such as infertility, a serious side effects making them unable to produce families [44,45]. Moreover, vaccine incidents that have occurred in the recent past have significantly decreased the public’s confidence in vaccines in China, especially among youngsters. However, it is believed that these incidents are less impacting among the older adults in terms of their previous positive experiences on other vaccines to themselves or families. They are also thought to have higher education, greater experience in healthcare settings, greater perceived vulnerability to COVID-19 infection, or higher overall medical and health risk profiles [44,46]. Significantly lower confidence and willingness were found among participants with higher education. However, no difference was detected in their vaccination coverage. We assume such phenomenon indicates the fact that they may have better access to more information and might be more suspicious, while they simultaneously have a higher awareness of the value of vaccine in preventing COVID-19.

By multivariate analysis, the confidence in vaccine safety was proved to be the most valuable factor in predicting willingness to vaccination for the whole sample, which is in accordance with previous studies [5,6,7,8,9,10,11]. However, different patterns were found among different HCWs. Different from the other four kinds of participants, public health employees seemed to care more about the vaccine effectiveness followed by their trust in government in formulating their willingness to get vaccinated. These results are probably due to their deep knowledge about the efficacy of vaccines in preventing related diseases. Confidence in vaccine safety correlated more closely with willingness among nurses, doctors, technicians and hospital administrators, also followed by their trust in government. This indicated their focus on vaccination safety as they might suppose the vaccine itself be a potential pathogen with side effects, even becoming sick after vaccination, and their natural immunity be strong enough to prevent them from COVID-19, in line with previous studies not only for COVID-19 vaccine [47]. Thus, future promotion strategies for vaccination among HCWs should be tailored to targeted populations.

In total 63.6% of the participants reported they had received the SARS-CoV-2 vaccine, which increased to 82.8% after adjusting for participants with medical conditions and contraindications. The current coverage means a high immunization coverage has been achieved among the HCW population in China [42]. A satisfactory coverage is supposed to be a direct reflection of HCWs’ high confidence and willingness rates, since China has initiated vaccination to COVID-19 among the priority populations in a voluntary way. However, we found 130 (7.1%) participants who expressed willingness to get vaccinated were still not vaccinated. In other words, willingness to get a vaccination does not imply necessarily actual uptake. Also, public health employees showed the highest willingness & vaccination rate and the lowest willingness & unvaccination rate. The medical technicians reported the highest willingness & unvaccination rate, followed by hospital administrator, nurses and doctors which were over two times more than that of public health employees. Issues like reports of sudden adverse events, negative mood before injection, schedule rearrangement, etc. may all lead to the potential reluctance to get vaccinated [48,49], especially among those who are suspicious with normal or low levels of vaccination confidence, which was supported by our results. Participants with willingness but without confidence reported a significant rate of unvaccination, compared to those who trusted in two or more dimensions of vaccination confidence. In other words, confidence might reflect a sustained momentum in promoting population’s vaccination to SARS-CoV-2. Thus, population’s confidence is not only the determinant to their willingness, but also guarantees their actual vaccine uptake.

## 5. Limitations

There are some limitations to our study. First, convenience sampling by online survey led to a biased selection, which may suffer from a lack of representativeness of the target population. The respondents were mainly female, urban residents, and people with a bachelor level education, which may somewhat restrict the generalizability of our findings. Second, although we performed quality control, there may be errors in the information because the online questionnaire cannot be modified after filling it out. Third, this study is a cross-sectional survey and causal relationships cannot be drawn. In order to dynamically assess vaccination willingness, confidence, and coverage of the Chinese population, a follow-up survey should be conducted to evaluate this issue.

## 6. Conclusions

Our findings have demonstrated a high level of willingness to receive vaccination among HCWs in mainland China, given their high-level confidence in SARS-CoV-2 vaccines, which is not only the determinant to their willingness, but also guarantees their actual vaccine uptake. In the meanwhile, different patterns of vaccine confidence and willingness were found among subgroups of HCWs. Vaccine coverage has been proved high among HCWs, however, reaching herd immunity among Chinese general population still need more efforts. HCWs are a key but not the only targeted population for vaccination. Specific and effective communication and health educational campaigns still need to be exerted to raise awareness regarding the necessity and benefits of vaccination against COVID-19 among different HCWs, since they are trusted sources of vaccine information.

## Figures and Tables

**Figure 1 vaccines-09-00993-f001:**
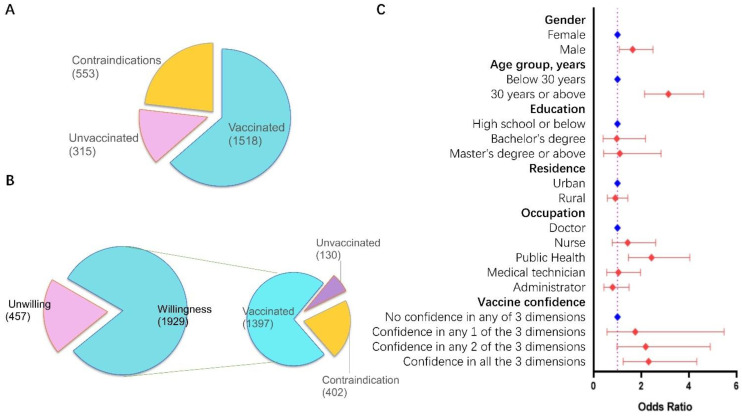
Prevalence and multi-factor analysis on vaccination willingness and coverage of the survey. Note: (**A**). The composition of the current vaccination status of the total population. (**B**). The proportion of vaccination uptake among those reporting willingness to vaccination. (**C**). Logistic regression analysis on vaccination uptake rate among those reporting willingness to vaccination. The reference groups were set as female, aged below 30 years, high school or below, urban residents, doctors for demographic characteristics and as no confidence in any of the three dimensions of vaccine confidence.

**Figure 2 vaccines-09-00993-f002:**
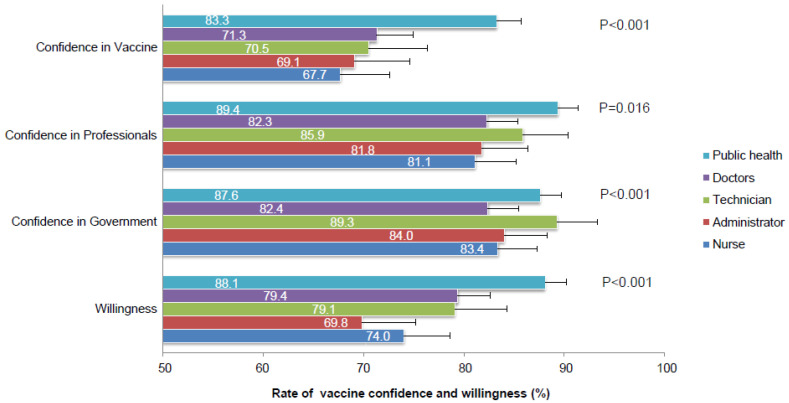
The rates of vaccine confidence and willingness among different HCWs.

**Figure 3 vaccines-09-00993-f003:**
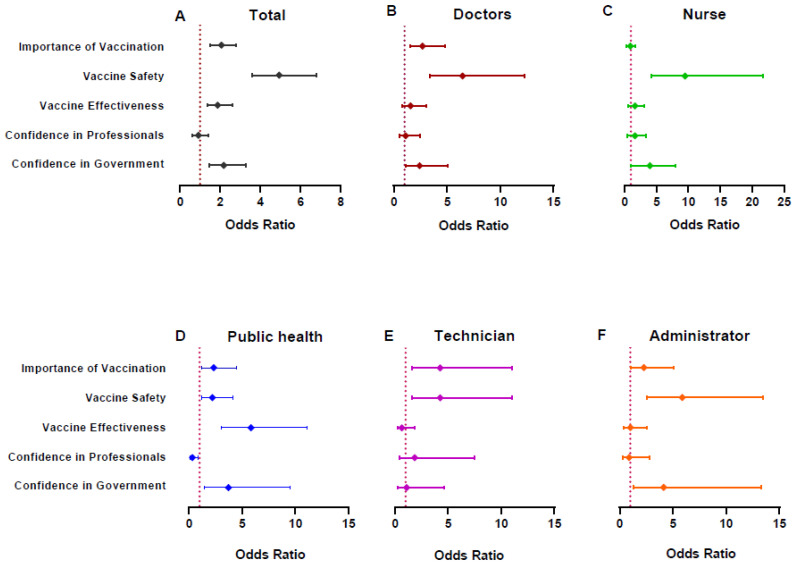
Analysis on the vaccination willingness by vaccine confidence within total sample and each HCWs subgroup basing on logistic regression model. Note: The coefficients of vaccination confidences shown in the graph were adjusted for gender, age group, education and residence. And the coefficients of demographic characteristics were not listed in the figure which could be found in Appendix A. For vaccine confidence, the reference groups were set as those reported negative confidence in each of the dimensions. (**A**). Analysis on the vaccination willingness by vaccine confidence within total sample. (**B**). Analysis on the vaccination willingness by vaccine confidence within doctors. (**C**). Analysis on the vaccination willingness by vaccine confidence within nurses. (**D**). Analysis on the vaccination willingness by vaccine confidence within public health employees. (**E**). Analysis on the vaccination willingness by vaccine confidence within technicians. (**F**). Analysis on the vaccination willingness by vaccine confidence within hospital administrators.

**Table 1 vaccines-09-00993-t001:** Characteristics of different healthcare positions.

Characteristics	Total *n* (%)	Doctor *n* (%)	Nurse *n* (%)	Public Health *n* (%)	Technician *n* (%)	Administrator *n* (%)	*p*
Gender							
Male	884 (37.0)	274 (45.4)	27 (7.7)	395 (42.7)	81 (34.6)	107 (38.9)	<0.001
Female	1502 (63.0)	329 (54.6)	323 (92.3)	529 (57.3)	153 (65.4)	168 (61.1)	
Age group, years							
below 30	621 (26.0)	144 (23.9)	129 (36.9)	197 (21.3)	77 (32.9)	74 (26.9)	<0.001
30 to 39	790 (33.1)	209 (34.7)	130 (37.1)	272 (29.4)	84 (35.9)	95 (34.5)	
40 to 49	627 (26.3)	170 (28.2)	67 (19.1)	278 (30.1)	44 (18.8)	68 (24.7)	
50 or above	348 (14.6)	80 (13.2)	24 (6.9)	177 (19.2)	29 (12.4)	38 (13.8)	
Education							
Below bachelor	160 (6.7)	29 (4.8)	28 (8.0)	63 (6.8)	13 (5.6)	27 (9.8)	<0.001
Bachelor	1684 (70.6)	380 (63.0)	321 (91.7)	646 (69.9)	157 (67.1)	180 (65.5)	
Master or above	542 (22.7)	194 (32.2)	1 (0.3)	215 (23.3)	64 (27.4)	68 (24.7)	
Residence							
Urban	1820 (76.3)	431 (71.5)	219 (62.6)	767 (83.0)	177 (75.6)	226 (82.2)	<0.001
Rural	566 (23.7)	172 (28.5)	131 (37.4)	157 (17.0)	57 (24.4)	49 (17.8)	
Total	2386 (100)	603 (25.3)	350 (14.7)	924 (38.7)	234 (9.8)	275 (11.5)	

## Data Availability

Original data are available on request. These were stored on password protected computers at Department of Laboratorial Science and Technology & Vaccine Research Center, School of Public Health, Peking University. Readers who wish to gain access to the data can write to the corresponding author.

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
