# Peer review of "Willingness and SARS-CoV-2 Vaccination Coverage among Healthcare Workers in China: A Nationwide Study"

_vaccines, 2021, doi:10.3390/vaccines9090993_

Round 1
Reviewer 1 Report
I was invited to revise the paper entitled "Willingness and Coverage of SARS-CoV-2 Vaccination on Healthcare Workers in China: A Nationwide Study". It was a cross sectional study that aimed to evaluate the willingness and coverage of COVID-19 vaccination among HCWs in China. The topic is interesting and it can improve the knowledge in this field.
I have some observations:
- Sample size estimation is lacking;
- Statistical analysis should be deeply described;
- Analyses reported in Figure 2 should be adjusted for age, gender and degree;
- Discussion should be improved. In particular differences in attitudes toward vaccination by age and HC position should be better described, as previously reported by 10.3390/vaccines8020248 and 10.1016/j.vaccine.2012.04.098. Also vaccine hesitancy towards other vaccinations should be discussed and compared;
- Comparison with previous similar studies should be discussed.
Reviewer 2 Report
This paper provides a clear and comprehensive overview of willingness and coverage of SARS-CoV-2 vaccination among healthcare workers in China.The study was well-designed and very meaningful. The revisions suggested below are mostly related to the methods and results.
Methods
1.This paper collected data byconvenience sampling method online. Could you add some evidence to prove the representativeness of the sample? For example, whether the sociodemographic characteristics(such as age, gender, education) of the sample were consistent with those of public statistics on national health care workforce?
2. Could you add some description of geographic or economic distribution of the sample to prove the national study?
Results
3. This paper included five working positions as HCWs (doctors, nurses, public health employees, medical technicians and hospital administrators). Public health employees was described as “CDC”in Table 1, while “public health” in Figure 1 and Figure2. It is better to use the same description.
Round 2
Reviewer 1 Report
Authors addressed all points. Paper is now acceptable for publication,